# Stress Granules in Infectious Disease: Cellular Principles and Dynamic Roles in Immunity and Organelles

**DOI:** 10.3390/ijms252312950

**Published:** 2024-12-02

**Authors:** Jaewhan Kim, Chang-Hwa Song

**Affiliations:** 1Department of Medical Science, College of Medicine, Chungnam National University, Daejeon 35015, Republic of Korea; jaewhan55@hotmail.com; 2Department of Microbiology, College of Medicine, Chungnam National University, Daejeon 35015, Republic of Korea

**Keywords:** stress granule, infection, infectious diseases

## Abstract

Stress granules (SGs) are membrane-less aggregates that form in response to various cellular stimuli through a process called liquid–liquid phase separation (LLPS). Stimuli such as heat shock, osmotic stress, oxidative stress, and infections can induce the formation of SGs, which play crucial roles in regulating gene expression to help cells adapt to stress conditions. Various mRNAs and proteins are aggregated into SGs, particularly those associated with the protein translation machinery, which are frequently found in SGs. When induced by infections, SGs modulate immune cell activity, supporting the cellular response against infection. The roles of SGs differ in viral versus microbial infections, and depending on the type of immune cell involved, SGs function differently in response to infection. In this review, we summarize our current understanding of the implication of SGs in immunity and cellular organelles in the context of infectious diseases. Importantly, we explore insights into the regulatory functions of SGs in the context of host cells under infection.

## 1. Introduction

Cells are continuously exposed to various internal and external stressors, such as changes in temperature, osmotic balance, nutrient, and pathogenic attacks. To maintain cellular homeostasis and activity under these stressful conditions, cells possess several stress responses [1]. These stress responses involve specialized mechanisms, such as organelle-specific stress pathways like endoplasmic reticulum (ER) stress [2,3] and mitochondrial stress [4,5]. One major response is the formation of stress granules (SGs), which are dynamic, membrane-less aggregates [6,7]. SGs consist of mRNAs, proteins, and other cellular components, formed via liquid–liquid phase separation (LLPS), and are classified as a type of ribonucleoprotein (RNP) granule [6,8,9,10].

SGs play an important role in maintaining cellular homeostasis, especially through their mRNA translation regulatory function [11,12,13,14]. Many of the RNAs and proteins found within SGs are associated with translation initiation, indicating that SGs regulate mRNA translation under stressful conditions [9]. However, because SGs are so dynamic, their composition and function depend on stress duration as well as the type of stress [15,16]. While SGs were initially characterized as an adaptation mechanism against non-infectious stressors, emerging evidence has highlighted their significant role in the immune response to infections [17,18].

Both viral and microbial infections can induce SG formation [19,20,21,22,23], though the underlying mechanisms of SG induction can differ depending on the type of pathogens and host cells [24]. SGs are involved in immune cell activity regulation and supporting or suppressing the host cell response against infection [25,26,27]. At the same time, SGs can also affect pathogen behavior by modulating infection dynamics, such as viral replication [28,29]. Given the transient and dynamic nature of SGs, SGs are normally disassembled when the stressor is removed [30], allowing cells to quickly conserve energy by temporarily halting unnecessary protein synthesis. However, in the case of infection-induced stress, where pathogens persist within host cells for extended periods, SGs play a more sustained role in modulating cellular responses and immune activity over time [15].

The formation of SGs shows distinct patterns in viral and bacterial infections, implying the various strategies pathogens use to react against a host’s defense system. During viral infections, SGs are rapidly induced as a part of potential innate immune response, driven by viral nucleic acid recognition [31]. This rapid SG assembly helps sequester viral components, limiting viral replication and the translation of viral proteins, while activating antiviral signaling pathways. In contrast, bacterial infections, including those caused by *Mycobacterium tuberculosis* (Mtb), often trigger more sustained SG formation through prolonged infection [32]. Persistent SGs may modulate the cellular activity of host cells, potentially suppressing overall protein synthesis and immune responses. Such prolonged SGs could have either a protective effect or an adverse effect on host cells. Understanding these pathogen-specific differences in SG dynamics and function is crucial for developing targeted therapeutic strategies against infectious diseases.

Understanding how SGs function in cells and how they can support or disrupt cellular activity and immune responses during infection has become an interesting research area for infectious diseases. Despite significant advances in this field, many questions remain about the regulatory mechanisms by which SGs affect immune responses and the onset of infectious diseases. This review aims to summarize the current understanding of SGs, highlighting its impact on immune cells and pathogens in the context of infectious diseases.

## 2. Principles and Properties of Stress Granules

### 2.1. Stress Granule Composition and Dynamics

SGs are dynamic, membrane-less aggregates that form in response to cellular stress. SGs consist of stalled preinitiation complexes, translation initiation factors [33,34], mRNAs [35,36], and scaffolding proteins [9], which are crucial for SG formation, function, and the regulation of cellular homeostasis during stress responses. At the core of SGs, RNA-binding proteins (RBPs) such as Ras GTPase-activating protein-binding protein 1 (G3BP1) and T-cell-restricted intracellular antigen-1 (TIA-1) recognize and bind to untranslated mRNAs, thereby forming stable core structures of SGs. These cores are surrounded by a less-dense shell that includes additional RBPs and translation initiation factors, which facilitate SG assembly and function [37,38,39]. It has been proposed that there are two models of SG formation [6]. First, in the ‘core first model’, clusters of ribonucleoproteins (RNPs) are formed and then grow larger and merge to form SGs with a dynamic shell. In the ‘LLPS first model’, phase-separated droplets form first, grow larger, and eventually concentrate to form the core of SGs [6].

One of the interesting features of SGs is their high dynamism, which allows SGs to actively exchange mRNAs and proteins through a less-dense shell. This dynamic property is important for adaptation to fluctuating intracellular or extracellular environments, and for the regulation of translation and the fate of mRNAs. Despite the dynamic nature of SGs, several mass spectrometry-based proteomic studies have provided valuable insights into the component of SGs, revealing a complex and variable protein landscape [40,41,42,43,44]. However, identifying SG components can be challenging due to the overlap of some proteins with other RNP granules, as well as variation in SG composition depending on the type of stressor, the duration of stress, and the specific cell types used. Additionally, from cell lysates, the isolation of SGs, which are non-membrane-bound structures, can be affected by artifacts during cell lysis, further complicating the identification process.

Thus, to study and identify SG components in response to specific stressors, it is essential to standardize key experimental factors. These include: (1) the type of stress stimulus, (2) the cell type or cell line used, (3) the duration of stress exposure, and (4) the method employed for SG isolation. Despite these challenges, SG proteomics provides critical insights into the molecular mechanisms by which SGs contribute to cellular stress responses and regulation, shedding light on their role in modulating translation and maintaining cellular homeostasis during stress.

### 2.2. Stress Granules and Cellular Functions Beyond Translation

Gene ontology (GO) biological process (BP) analysis of the SG proteome reveals that SG components are predominantly involved in mRNA processing and translation (Figure 1). SGs are well-known for suppressing protein translation during stress and helping cells conserve energy and adapt. However, recent studies indicate that mRNAs within SGs can still undergo the full translation cycle—initiation, elongation, and termination [14]. This suggests that the high local concentration of translation machinery in a dense environment of SGs may promote translation under specific conditions, although further research is required to understand the underlying mechanisms, including how amino acids and tRNAs dynamically travel between SGs and the cytoplasm.

Interestingly, GO cellular component (CC) analysis shows that SGs also contain proteins from various organelles and cell membranes, including a significant number of proteins categorized within P-bodies (PBs) (Figure 2). These proteins are believed to play a role in the regulation of specialized functions in each organelle, such as mitochondrial ATP production or unfolded protein responses from the endoplasmic reticulum (ER). Notably, PBs-associated proteins are also found in the SG proteome, evidenced by several studies that demonstrate the communication between SGs and PBs [45,46]. Additionally, SGs are linked to cellular energy metabolism, as they contain proteins involved in mitochondrion function (Figure 2). Recent studies show that SGs interact with mitochondria and mitochondria can also form SGs, supporting the idea that SGs may influence energy metabolism during stress [47,48,49]. Focal adhesion-associated proteins are sequestered into SGs (Figure 2). One of the SG markers, the receptor of activated protein C kinase 1 (RACK1) integrates the adhesion, polarity and motility of cells, affecting the growth and survival of the cells [50,51]. At the same time, a study showed that focal adhesion kinase (FAK) is localized in SGs in mouse embryonal carcinoma cell line P19 cells, but not in focal contacts, as seen in other cell lines [52]. Most recently, either anoikic stress or loss of adhesion has been shown to induce the formation of SGs via the inhibition of FAK or the loss of adhesion signaling [53].

GO molecular function (MF) analysis reveals that protein- and RNA-binding-associated proteins are dominantly sequestered in SGs, but proteins involved in ATP hydrolysis, cadherin binding, and chaperone activities are also present (Figure 3). However, research on the roles of these proteins in SGs remains limited and needs further exploration.

Comprehensively, among the five previously published stress granule (SG) proteome lists, only nine proteins were consistently identified across all datasets (Figure 4; DDX3X, EIF3B, EIF3E, EIF3F, EIF3L, EIF4A1, EIF4A1, EIF4G1, PABPC4, and PCBP1) [40,41,42,43,44]. SG proteome list 1 contains many proteins involved in mRNA splicing and nuclear transport. On the other hand, list 2 features proteins performing classical SG functions, such as RNA sequestration. Similarly, list 3 is enriched with proteins clustered into the R3H nucleic acid-binding domain. List 4 contains well-known SG components involved in general SG formation and maintenance. Finally, list 5 shows enrichment for proteins associated with mitochondrial and ER functions, suggesting SGs may have broader roles beyond translational regulation. These findings underscore that SG composition is highly dynamic and influenced by various factors, including the stress environment, cell type, type of stress, and SG isolation methods. In the context of infection-induced SGs, factors such as the number and type of pathogens, infection duration, and immune cell type can significantly impact SG composition. Despite these complexities, no comprehensive omics studies have yet analyzed SGs formed during infections. Given that infections introduce more variables compared to typical stressors like oxidative stress or heat shock, future studies must carefully standardize experimental conditions. Establishing these parameters will be essential to accurately define and understand the role of SGs in pathogen infections.

Beyond translation regulation, SGs are involved in a range of cellular functions, including the modulation of signaling pathways and stress responses. SGs interact with PBs to facilitate mRNA degradation or recycling, depending on the cellular needs [54]. SGs also recruit signaling proteins such as kinases, phosphatases, and stress response components, creating a hub for cellular stress signaling. Additionally, SGs have been implicated in the regulation of apoptosis, where their formation may delay cell death under mild stress [51,55]. However, if stress persists, SG components can be actively changed to promote pro-apoptotic signaling, helping cells balance survival and adaptation with the initiation of cell death when damage is irreparable.

### 2.3. Stress Granules and Immune-Related Proteins

SGs play a crucial role between cellular stress responses and immune defense, particularly during viral infections [56,57]. When viral RNAs are detected, SGs can inhibit viral replication by isolating the viral RNA with key translation machinery [58,59]. Immune-associated proteins, such as pattern recognition receptors, retinoic acid-inducible gene I (RIG-I) and melanoma differentiation-associated protein 5 (MDA5), are localized into SGs and further activate antiviral signals, such as type I interferon (IFN) response, and sense viral RNA [60,61]. In addition to their antiviral function, SGs can also modulate inflammatory signals by recruiting components of the inflammation pathway, including nuclear factor kappa B (NF-κB) [62,63].

The enrichment of innate immune response-related biological processes within the SG proteome further highlights SGs’ active role in immune defense (Table 1). Proteins involved in viral RNA recognition, interferon signaling, and inflammatory responses are abundant in SGs, suggesting that SGs are directly involved in pathogen control and immune response. This involvement of immune-related proteins indicates that SGs are not only focused on mRNA regulation but also serve as central hubs for immune signaling during stress conditions.

## 3. Stress Granules and Organelles in Infected Cells

The dynamic interactions between SGs and organelles such as ER, mitochondria, and lysosomes, highlight their regulatory roles in cellular stress responses (Figure 5). The ER serves as a platform for SG assembly, facilitates their division through ER tubules, and exchanges mRNA with SGs to regulate translation and stress adaptation through ER stress regulation by clustering inositol-requiring enzyme 1 (IRE1α) [64,65]. SGs also modulate mitochondrial activity by redirecting fatty acids to lipid droplets, inhibiting β-oxidation [48]. More directly, in mitochondria, SGs interact with MAVS to enhance antiviral defense and influence the mitochondrial unfolded protein response (UPR_mt_) [63,66]. Additionally, SGs repair damaged lysosomal membranes and hitchhike on lysosomes for intracellular transport [67,68]. These findings emphasize that SGs are not isolated cellular components but highly dynamic components that communicate with other organelles. Because pathogen infection influences the activity and normal function of various organelles of host cells [69,70], the communication between SGs and organelles could fine tune the appropriate activity of specific organelles, and thus interactively cope with the infections. Understanding these interactions is important, particularly in the context of pathogen-induced SGs, as it may reveal novel insights into how cells cope with infections and maintain cellular homeostasis upon infection. Future studies should aim to understand SG–organelle communications to reveal their regulatory roles in cellular activity and disease mechanisms.

### 3.1. Stress Granules and Endoplasmic Reticulum

ER plays a critical role in maintaining the protein life cycle, including protein synthesis, folding, and secretion. In addition, ER communicates with other membrane-bound organelles, acting as a networking platform to regulate their activity (Figure 5). In 2020, a study demonstrated that the shape of ER—ER tubules and cisternae—regulates the formation and division of membrane-less RNP granules, by which ER tubules promote PB formation and serve as sites of PB and SG dynamic changes [65]. Another study emphasizes the importance of ER–SGs communication for a central site of SG assembly, determining whether mRNA is translated or not [71]. These studies provide insights into how membrane-less aggregates like SGs are formed and dynamically exchange mRNAs with the ER [72]. More recently, it has been shown that ER regulates SG formation and provides mRNAs to SGs, and ER stress response is regulated by SGs. It was suggested that the clustering of IRE1, a key protein in the ER stress response, occurs through phase separation, similar to the basic assembly mechanism of SGs, using SGs as a clustering platform [73]. When SG formation is inhibited, ER stress responses—XBP1 splicing and the pro-survival pathway—are impaired, indicating that SGs can regulate part of the ER’s function.

Various pathogen infections induce both ER stress and SG formation to host cells [74,75,76]. Because these two responses occur together during infection, it is theoretically thought that SG formation helps host cells to adapt and survive infection stress. However, persistent SGs and ongoing ER stress during infection can have adverse effects on host cells [15]. The function of ER and its stress response in infected cells likely depends on the dynamics of SG assembly and disassembly. Because SG components might be diverse depending on the type of infectious agent, further study is needed to explore the relationship between ER stress and SGs across different pathogens.

### 3.2. Stress Granules and Mitochondria

SGs can modulate the activity of mitochondria by regulating the mitochondrial permeability of fatty acids via VDAC porin modulation (Figure 5) [48]. More directly, it has been shown that one of the SG components, NUDT2, can regulate the activity of mitochondria by allowing IPS-1, residing in mitochondrial outer membrane, to be recruited into SGs [77]. Another study has shown that SGs and mitochondrial unfolded protein response (UPR_mt_), which are important for maintaining mitochondrial homeostasis, are interwound with each other [66].

During infections, cells require substantial energy [78,79]. The infected cells fine tune the activity of mitochondria to supply an adequate amount of energy to meet energy requirements [80]. In addition to energy production, mitochondrial-mediated immuno-regulatory functions are an important defense system against infection [81]. Mitochondria are actively involved in innate immune activity, producing reactive oxygen species (ROS) and activating pattern recognition receptors such as mitochondrial antiviral-signaling protein (MAVS) to defend against viral and microbial infections [82,83].

However, pathogens may also interfere with the normal immune activity of host cells by regulating the mitochondrial dynamic and innate immune activity of host cells [84]. A study showed that infection-induced cell death is inhibited by the increased mitochondrial energy metabolism [85] via mammalian target of rapamycin complex 1 (MTORC1), which is an SG component [86]. These two studies imply that SGs are one of the survival mechanisms of pathogens by which SGs sequester MTORC1, inhibiting their normal functions in mitochondrial activity regulation.

### 3.3. Stress Granules and Lysosome

Lysosomes are important organelles for intracellular defense against infected pathogens [87]. However, many pathogens can induce lysosome rupture, triggering mitochondrial protein degradation and cell death to evade the immune responses of host cells [88]. It has been suggested that the inhibition of lysosomal activity could alter SG morphology and composition [89]. Before the ER was recognized as crucial for SG formation, a study in 2019 highlighted the relationship between SGs and lysosomes. The author showed that SGs utilize ANXA11 to hitchhike on lysosomes for long-distance transport (Figure 5) [68]. This finding provides insights into how SGs can be delivered to distal parts of cells like neurons, though further study is necessary to demonstrate which way is dominant—whether SGs form directly in distal regions of cells or are transported via lysosome hitchhiking.

Following this research, another study revealed that SGs are induced by lysosome-damaging stimuli, such as infections and proteopathic tau [32]. Importantly, a recent study demonstrated that SGs stabilize ruptured lysosomes, facilitating recovery through both ESCRT-dependent and independent mechanisms [67]. This study underscores the importance of SGs in infection-induced stress, as inhibiting the formation of SGs allows pathogens to exploit lysosome damage, thereby increasing infection vulnerability.

The idea of the communication between SGs and lysosomes is further supported by a study showing that upon lysosomal damage, eIF2α is phosphorylated to initiate SG formation [90]. This suggests that SGs may help the lysosome repair system and host cell survival under infection-induced stress. Taken together, SGs may act as a recovery mechanism when lysosomes are compromised by infection, though their role likely varies depending on the specific pathogen involved.

## 4. Stress Granules and Immunity

### 4.1. Stress Granules and Innate Immunity

#### 4.1.1. Platforms for Immune Signaling Pathways

SGs could act as a signaling center by recruiting immune receptors and adaptors involved in antiviral pathways. During viral infections, SGs sequester viral RNAs along with immune receptors, such as RIG-I and MDA5, which recognize foreign viral RNAs and regulate type I IFN production [61,91]. By localizing immune sensors in SGs with high local concentration, cells are able to efficiently activate IFN responses and amplify downstream pathways, enhancing antiviral defenses. Furthermore, SGs interact with mitochondria, enabling SG-resident proteins to communicate with MAVS, which amplifies the immune response [92,93]. These studies highlight the function of SGs as potential coordinators of innate immune responses.

#### 4.1.2. IFN Response

SG formation is strongly enhanced by IFN treatment in adenosine deaminase acting on RNA 1 (ADAR1)-sufficient cells, infected with measles virus (MV) [94,95]. Type I IFN-induced SG formation occurs via STAT1 and STAT2 pathway in response to canonical IFN signaling. This study showed the connection between IFN responses and SGs, highlighting ADAR1’s role as a suppressor of IFN and SG responses [96].

A RNAi screen identified nearly 100 host genes associated with RIG-1 mediated IFN production, including the SG scaffolding protein G3BP1. G3BP1 enhances RIG-I-induced IFN-β mRNA synthesis, binds directly to RIG-I, and colocalizes with RIG-I and viral RNAs. This finding suggests G3BP1 as a key factor of RIG-I signaling, potentially serving as a sensor to boost RIG-I recognition of viral RNA [61].

Growth arrest and DNA damage-inducible beta (Gadd45β) binds to G3BP1, enhancing its RNA-binding affinity and conformational expansion, thus promoting SG assembly and facilitating interferon signaling [97]. The authors show that Gadd45β acts as a positive regulator of type I IFN responses by targeting G3BP during RNA virus infection. Loss of Gadd45β disrupts SG formation and interferon signaling, leading to impaired cytokine production.

#### 4.1.3. Dual Role of SGs in Immune Modulation and Pathogen Evasion

While SGs contribute to innate immunity, pathogens have also evolved mechanisms to disrupt SG formation and evade immune responses. Several viruses, such as herpes simplex virus (HSV) and poliovirus, rapidly replicate by inhibiting the formation of SGs which sequester viral components needed for replication [98,99]. In addition, some bacteria and viruses hijack SG-associated proteins, manipulating the function of SGs for their own benefit or avoiding the immune surveillance of host cells [21,57,100]. Understanding these interactions will offer valuable insights into host-pathogen dynamics and potential therapeutic strategies for managing the role of SGs, thereby enhancing innate immune defenses against infection.

### 4.2. Stress Granules and Adaptive Immunity

#### 4.2.1. Antigen Presenting Cells

During infection, efficient antigen presentation by antigen presenting cells (APCs) is important for the activation of adaptive immune responses [101]. SGs could affect this process by regulating the translation of mRNAs that code for major histocompatibility complex (MHC) molecules and co-stimulatory signals for T cell priming [102]. This allows SGs to modulate antigen processing pathways in infected cells, affecting the efficiency of pathogen-derived antigen presentation to T cells. Although it is still elusive whether SGs are important factors during innate antiviral immune response [103], SGs might help infected cells fine tune the degree of antigen presentation, which can either enhance or reduce T cell activation depending on the requirements for a balanced immune response of host cells [17].

#### 4.2.2. T and B Cell Activation

SGs modulate T cell responses by regulating the translation of cytokine mRNAs essential for T cell proliferation and survival during infection-induced stress [104]. For instance, cytokines like IL-2, necessary for T cell proliferation, are controlled at the translational level within SGs, which affects IL-2 production and the subsequent expansion of T cells in response to infection [105]. This regulation accomplished by SGs ensures that T cells can respond appropriately upon infection, preventing excessive inflammation. Through this regulatory role of SGs, SGs may support the adaptive immune system to modulate the intensity and duration of T cell responses in the presence of persistent or chronic infections, where uncontrolled excessive T cell activation could otherwise damage the host [106].

As yet, there is no study about the relation between SGs and B cell activation. One study on mendelian susceptibility to mycobacterial disease (MSMD) showed that individuals deficient in zinc finger NFX1-type containing 1 (ZNFX1) protein, which is an SG component, have normal proportions of B cells among peripheral blood monocytes (PBMCs), compared to controls [107]. As this study is based on the single protein, ZNFX1, not SG itself, it is still necessary to determine whether SGs would regulate the B cell population of activation.

## 5. Stress Granules and Infectious Diseases

### 5.1. Viral Infection and Diseases

When viral infection is established, a virus is replicated using the host cells’ translation machinery to produce viral capsid and viral nucleic acids [108]. In this process, the exposed viral nucleic acids in cytoplasm are detected by several proteins of host cells, resulting in the initiation of immune responses [109]. As the basic structure of SGs are protein–protein, RNA–RNA, and protein–RNA interactions, viral nucleic acids can serve as SG forming-materials [110]. In addition, viral infection induces several cellular stress responses, such as the activation of protein kinase R or the inhibition of eIF4G and eIF4A to initiate SG assembly [99,111]. Through this, a virus can inhibit immune responses and enhance viral replication by increasing the local concentration of viral nucleic acids and translation machinery; however, other types of viruses uncouple the cellular stress signal and SGs (Table 2) [112]. By suppressing SG formation, they prevent viral RNA sequestering into SGs [113]. TIA-1/TIAR protein binds with viral RNA, losing its role in SG assembly, to help viral replication and increase apoptosis for dissemination [114].

Foot-and-mouth disease virus (FMDV) inhibits SG assembly to facilitate its replication. By degrading SG components such as G3BP1, FMDV ensures viral mRNA is readily available for translation, bypassing the host’s immune defenses [115]. FMDV employs proteases to cleave SG scaffolding proteins, G3BP1 and G3BP2, suggesting how viruses evade immune defense mechanisms to establish infection [116].

Zika virus (ZIKV) modulates SGs to avoid the immune activity of host cells, allowing viral RNA translation to be continued. ZIKV interferes with SG-associated proteins, enabling its replication and contributing to pathogenesis [21,117]. However, ZIKV not only suppresses SG formation but also exploits SG-associated pathways to enhance its replication within host cells. By sequestering SG proteins required for replication, G3BP1 functions to increase viral replication [118].

Ebola virus (EBOV) targets SG dynamics to evade host immunity. EBOV proteins block SG formation, ensuring viral RNA translation and efficient replication [119]. This interaction helps EBOV evade host defenses as a part of survival mechanism within immune cells [120]. This survival mechanism, by which EBOV inhibits the formation of SGs, is supported by a study showing that certain SG proteins are sequestered into EBOV inclusion, not SGs [121].

Influenza A virus (IAV) can inhibit SG formation through its non-structural protein NS1, enabling viral RNA to avoid sequestration [122]. By countering SG assembly, IAV enhances replication efficiency and evades host immune responses, contributing to its infectious potential [123]. These studies provide understanding of how IAV disrupts SG-mediated antiviral defenses. In addition to inhibiting SG formation, IAV also modifies the host cell environment to favor viral mRNA translation over host gene expression [124]. By inhibiting SG components involved in antiviral signaling, such as PKR and G3BP1, IAV reduces the cellular stress response, allowing for efficient viral protein synthesis [62,125]. This strategy not only enables rapid viral replication but also disrupts innate immune responses.

SARS-CoV-2 is able to remodel SGs to atypical foci to suppress innate immunity using a nucleocapsid protein [126]. The nucleocapsid protein, SARS-CoV-2 N protein, sequesters two key scaffolding proteins of SGs, G3BP1 and G3BP2, leading to the inhibition of SG formation. In addition, capsid protein also binds host mRNAs to alter the post-transcriptional program required for proper host stress response [127]. This strategy underscores the importance of SG modulation in the SARS-CoV-2 infection-replication cycle and suggests that SG-targeting therapies may offer new treatment options.

### 5.2. Bacterial Infection and Diseases

Bacterial infections often trigger SGs as part of the host’s stress response to contain bacterial growth (Table 2). While SGs can help control infections by sequestering proteins and mRNAs involved in immune signaling, many bacterial pathogens disrupt SG formation through specific effectors. This interaction allows bacteria to evade or manipulate the host immune system, which can lead to more severe infections [128]. During infection, SGs serve as signaling platforms for immune responses, recruiting proteins involved in cytokine production and inflammatory pathways, which are crucial for pathogen clearance [129]. The modulation of SG dynamics by pathogens also suggests that SGs may contribute to host cell fate, where bacterial evasion of SGs can lead to programmed cell death or persistence, aiding in bacterial survival within the host [23,130].

*Salmonella* is the first identified microbe which causes host SG formation [131]. *Salmonella* species are known to induce SG formation in host cells, initiating a stress response to limit bacterial growth [22]. *Salmonella*, however, produce virulence factors that can disassemble RNP granules, countering host defenses. For instance, studies show that *Salmonella* effectors inhibit SG formation by targeting proteins like G3BP1, essential for SG assembly, enabling bacterial replication within host cells [132,133]. Although the function and role of SGs upon *Salmonella* infection are still elusive, they are likely to work as positive regulators of immune activity.

Upon *Shigella* infection, SGs initially form to sequester host translational machinery, limiting bacterial replication. They use a type III secretion system (T3SS) to deliver proteins that initiate the formation of SGs [22]. Interestingly, *Shigella* infection induces the phosphorylation of eIF2-α and SG formation to a limited extent, still they have a mechanism to inhibit the formation of SGs [23].

Pathogenic *E. coli* strains, causing intestinal infections, induce SGs as well. For example, Shiga-toxin-producing *E. coli* (STEC) strains trigger SG formation by sequestering essential host proteins and mRNAs [134]. However, some *E. coli* strains, such as enterotoxigenic *E. coli* (ETEC) cannot induce the formation of SGs in Caco-2 cells. Interestingly, *E. coli*-infected Caco-2 cells were not able to induce SG assembly, due to the impaired phosphorylation of eIF2α [135].

The intracellular pathogen *Listeria* can induce SG formation upon host cell invasion with elevated levels of phospho-eIF2α [136]. Like other bacterial infection-induced SGs, the function of *Listeria* infection SGs is largely unknown. Given that *Listeria* is an intracellular pathogen, which should continuously modulate the immune responses of host cells, they consistently introduce cellular stress to the cells, inducing SGs. Those persistent SGs are likely to help bacteria to survive within the host cells, by which SGs suppress immune responses.

Another intracellular pathogen, *Mycobacterium tuberculosis* (Mtb) infection triggers oxidative, ER and inflammatory stresses in macrophages that may lead to SG [137,138]. Mtb infection also induces lysosomal damages, which can induce SG assembly [32,66]. The underlying mechanism of SG induction upon Mtb infection is still unclear because integrated stress responses, including ER stress, oxidative stress, and inflammatory stress, are all elevated in Mtb-infected cells.

*Helicobacter pylori* (*H. pylori*) infection triggers a variety of cellular stress responses in a host’s gastric epithelial cells. These responses include oxidative and inflammatory stresses, potentially inducing stress granule (SG) formation [139,140]. However, direct evidence of the role or formation of SGs during *H. pylori* infection remains still elusive. *H. pylori*’s virulence factors (e.g., CagA protein) are known to manipulate host cell pathways, including inflammatory and apoptosis, which might intersect with SG dynamics [141]. Further research is necessary to clarify how *H. pylori* affect SG assembly and whether it uses this mechanism to evade immune responses or enhance survival.

### 5.3. Differences in Viral and Bacterial Stress Granules

The formation of virus-induced SGs is initiated with the detection of viral nucleic acids in cytoplasm, and the downstream stress signal is activated [111,142]. The type of virus determines whether the SGs are helpful to the host cells or to the virus [125]. Intracellular bacterial infections are generally established by phagocytosis and extracellular bacteria also stimulate cells by secreting toxins [143]. During this process, various stress signal pathway responses, such as a rapid increase in ROS, ER stress, energy depletion and cAMP increase, are triggered [129]. Like viruses, bacterial DNA can be released into cytoplasm, which might affect SG assembly [144,145]. However, so far, it is largely unknown whether released and leaked nucleic acids of bacteria manipulate SG dynamics. It is known that SGs are formed by integrated stress responses upon bacterial infection, but the impact of bacteria-induced SGs on bacterial proliferation and pathogenesis has not been studied until now.

Differences depending on the virus and bacteria suggest that SGs can play various roles in terms of immunity and understanding the properties and components of each SG in relation to various pathogens would provide deep insights into the prevention and treatment of infectious diseases.

**Table 2 ijms-25-12950-t002:** Pathogen modulation of stress granules.

Pathogens	Effect on SGs	Key Viral Factors	Target	Mechanism	Outcome	Reference
Influenza A Virus	SG suppression	NS1 protein	PKR, G3BP1, eIF4G	Inhibits SG formation by preventing PKR activation and RNA sequestration	Enhances viral replication, evades immune responses	[122,123]
SARS-CoV-2	SG suppression and remodeling	Nucleocapsid protein	G3BP1, G3BP2	Remodels SGs, sequesters G3BP proteins	Suppresses innate immunity, enhances viral replication	[91,126,127]
Foot-and-Mouth Disease Virus	SG suppression	L and 3C proteases	G3BP1, G3BP2	Degrades SG scaffolding proteins	Ensures viral mRNA translation, bypasses host defenses	[116]
Zika Virus	SG exploitation	Capsid protein	G3BP1, TIA-1	Uses SG proteins to enhance replication	Facilitates viral replication within host cells	[21,117,118]
Ebola Virus	SG suppression	Viral protein (VP35)	SG-associated proteins	Blocks SG formation, sequesters SG proteins	Prevents immune detection, enhances viral replication	[119,120,121]
Poliovirus	SG disassembly	3C protease	Cleaves G3BP1 and eIF4G	Disassembles existing SGs	Enhances viral RNA translation	[146]
Herpes Simplex Virus	SG inhibition	Endoribonuclease VHS	eIF2α dephospho-rylation, PKR	Prevents SG formation	Maintains host translation machinery for viral replication	[98,147]
*Salmonella enterica*	SG disassembly	Unknown	G3BP1, PKR	Inhibiting PKR	Evades host immune response, promotes intracellular survival	[87,131,133]
*Mycobacterium tuberculosis*	Persistent SG induction	Unknown	eIF2α phosphory-lation, lysosomal damage	Induces persistent SGs via ISR and endolysosomal damage	Repairs damaged lysosome, helps or suppresses bacterial survival within macrophsges	[32,67]
*Listeria monocytogenes*	SG induction and persistence	Unknown	eIF2α phosphory-lation	Induces SG formation during host invasion	Modulates host stress response to help survival	[136]
*Shigella flexneri*	Limited SG induction	Type III secretion system	eIF2α phosphory-lation	Induces transient SG formation, then inhibits assembly	Manipulates host immune response, facilitates bacterial spread	[22,23]
*Escherichia coli*	SG induction	Shiga toxins	Ribosome-inactivating proteins	Induces SG formation by halting translation	Sequesters host proteins, promotes bacterial survival	[134,135]
*Helicobacter pylori*	Unknown	Peptidoglycan	eIF2α phosphory-lation	Potentially induces SG formation	Unknown	[139,140]

## 6. Prospects and Future Directions

Because host-directed therapy (HDT) has emerged as a novel approach in the field of infectious disease treatments, it is important to study how cellular activities, including immune reactions and energy metabolism are regulated [148]. Viral and bacterial infections initiate immune cell activity, maintain the appropriate level of immune cell activity, and enable cells to overcome infection. However, infection itself is also stressful for cells, which results in the formation of SGs. SGs, which have mRNA and proteins that play various roles, have highly dynamic features, so their roles and functions differ depending on which pathogen is infected. Therefore, it may be the dynamic characteristics of SGs that cause different onset of infectious diseases and different immune activities. Studies on the role of SGs caused by viral infection are relatively well-known, but the role of SGs caused by bacterial infection is still largely unknown. Furthermore, it is unclear what effects SGs can actually have on patients with infectious diseases. SGs can play a role in the function of tissue or organs as SGs can manipulate uninfected bystander cells as well as infected cells. Future research strategies should aim to comprehensively analyze and elucidate the characteristics and functions of SGs in diverse immune cell types and pathogens. This will provide critical insights into how the dynamic properties of SGs influence immune cells during infections. Furthermore, it is essential to explore the integrated roles of SGs in cellular activities beyond immune responses. Such efforts could ultimately pave the way for fundamental solutions to numerous intractable diseases, offering new avenues for therapeutic innovation.

### 6.1. Stress Granules in Host-Directed Therapy

SGs may play a different role depending on the pathogen infected—inducing SGs as a part of defense mechanism or to subvert host immune responses. Future studies should explore how various pathogens influence SG function, including assembly and disassembly, SG composition, and mRNA translation regulation. This would provide insights into pathogen-specific therapeutic strategies. Despite the promise of HDT in treating infectious diseases, the role of SGs in modulating the efficacy of HDT remains largely unknown. It should be investigated how manipulating SG dynamics could enhance or hinder the effectiveness of HDT as SGs are largely involved in various cellular activity upon infections. Whether promoting SG formation or enhancing SG disassembly could be used to modulate immune responses, apoptosis, and the autophagy pathways of host cells should also be addressed, as these are crucial HDT strategies.

### 6.2. Temporal Dynamics of Stress Granules During Infection

The composition of SGs is dynamically altered across the SG life cycle [16]. Thus, SG formation and resolution likely vary across different stages of infection [15]. Early in infection, SGs may be host-beneficial, inhibiting pathogen replication and undergoing integrated stress responses to cope with infection [31]. In contrast, prolonged SGs might suppress overall cellular activities, including protein synthesis, energy production, and immune responses, and facilitate pathogen survival [15]. In addition, in order to exhibit an appropriate immune response against infection, the activity and maturation of immune cells must occur at the proper time. Given the role of SGs in translation and organelle regulations, SGs may be important coordinators of immune cell activation or maturation [104]. Understanding these duration-dependent dynamics of infection in innate versus adaptive immunity is essential to elucidate the pathophysiological role of SGs in each infectious disease.

## Figures and Tables

**Figure 1 ijms-25-12950-f001:**
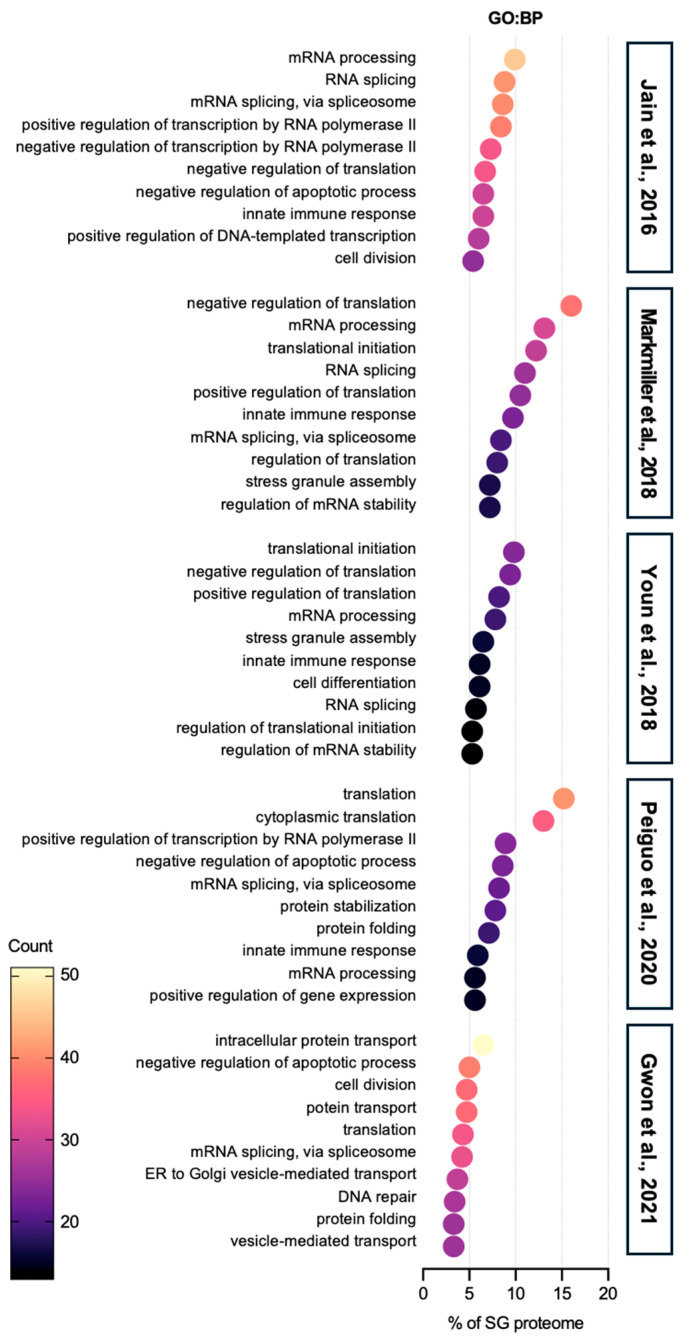
Gene ontology: biological process analysis of SG proteome. The top 10 categories are shown from GO analysis on biological process (GO:BP) of proteins in SGs from five SG proteomes studies [40,41,42,43,44]. All five SG proteomes have RNA-related categories, such as RNA splicing and translation, in the top ranks. In the category of proteins related to translation, both negative regulation of translation and positive regulation of translation terms are found. In addition to the RNA-associated terms, proteins, which are involved in basic cellular activities such as cell division, apoptosis, and protein transport, are abundant in SG proteomes. In terms of immunity, SG proteomes contain a certain list of proteins associated with immunity.

**Figure 2 ijms-25-12950-f002:**
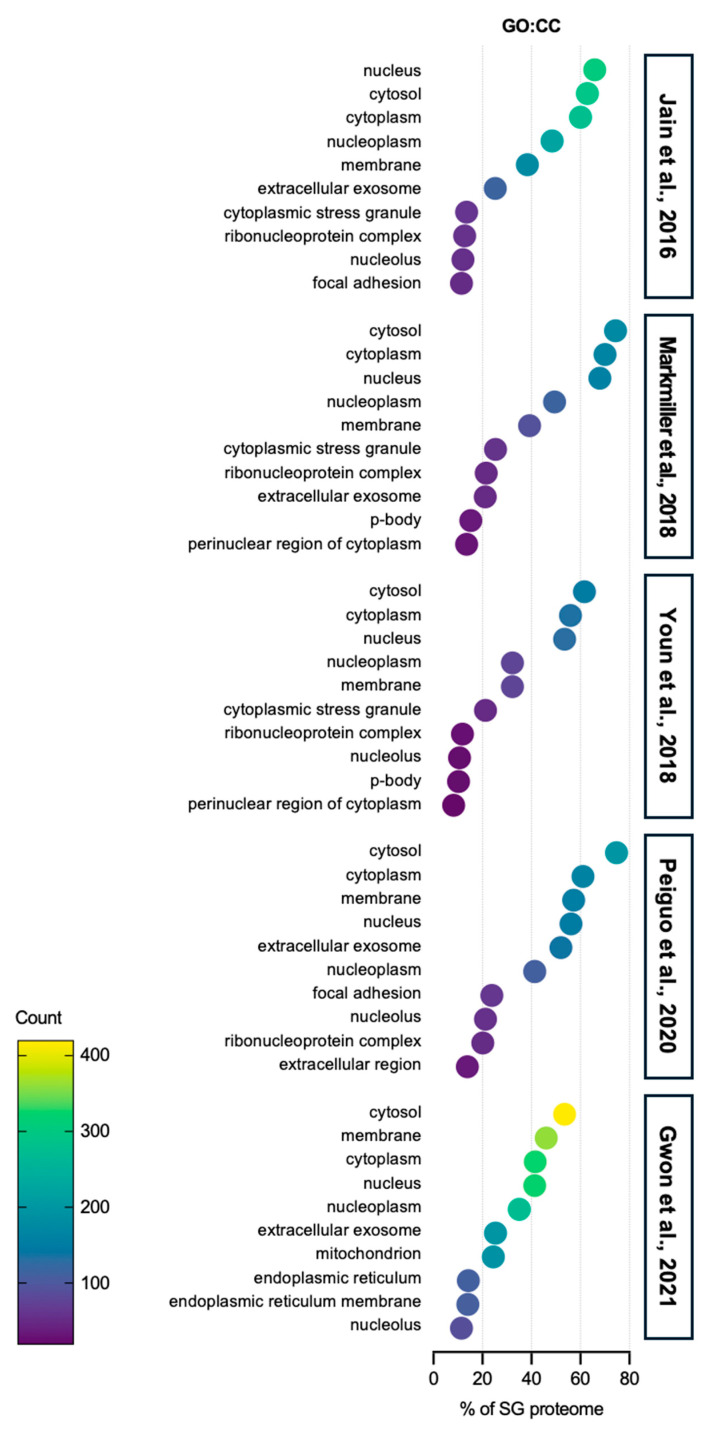
Gene ontology: cellular component analysis of SG proteome. The top 10 categories are shown from GO analysis on the cellular component (GO:CC) of proteins in SGs from five SG proteomes studies [40,41,42,43,44]. Over 50% of SG proteins are related to the cytoplasm. Proteins associated with exosomes, which are secreted out of the cells, also account for more than 20% of SG proteins from five SG proteomes. Most interestingly, proteins from other organelles, such as P-body, endoplasmic reticulum, and mitochondria, are also sequestered into SGs, indicating that SGs can actively regulate the activity of specific organelles in stress conditions.

**Figure 3 ijms-25-12950-f003:**
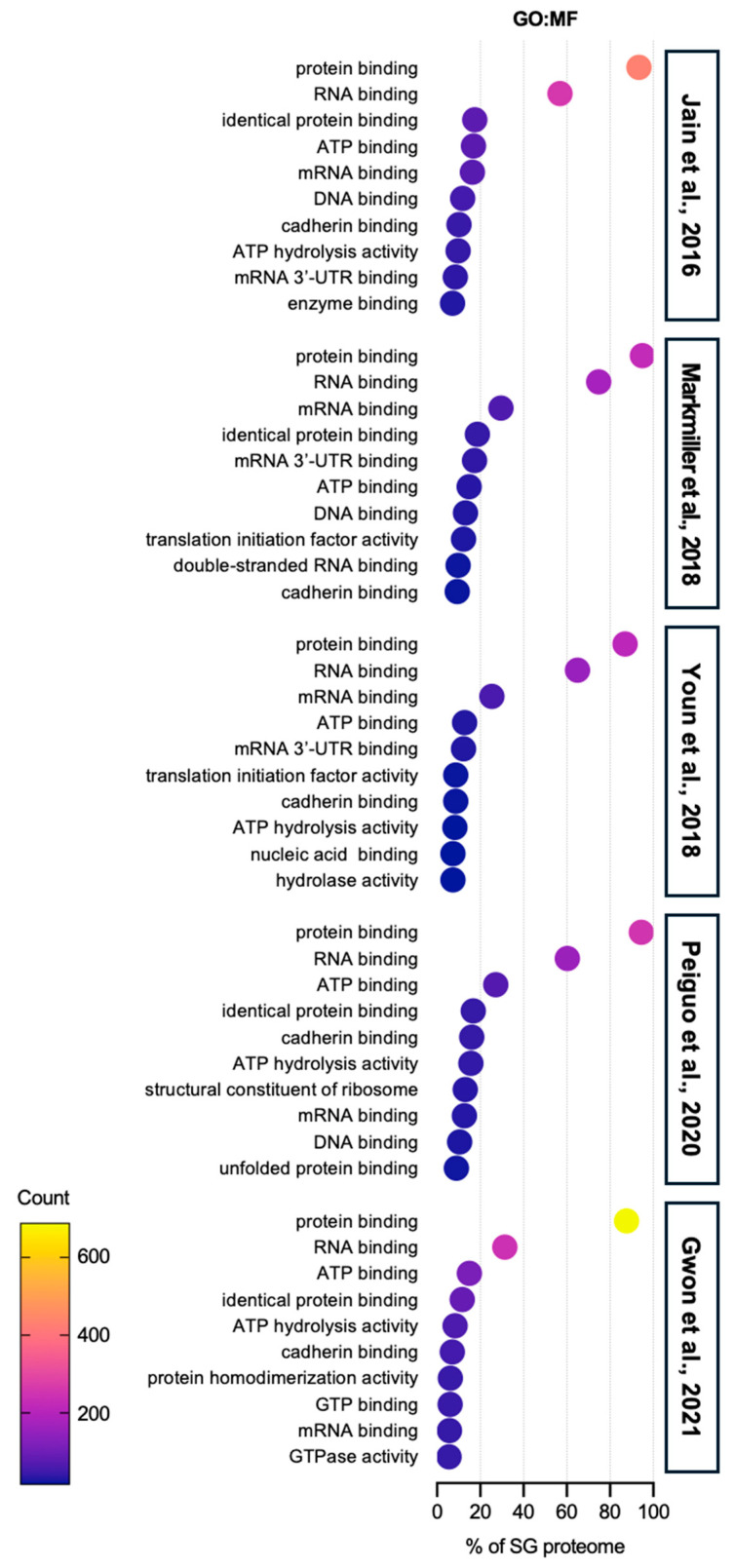
Gene ontology: molecular function analysis of SG proteome. The top 10 categories are shown from GO analysis on the molecular function (GO:MF) of proteins in SGs from five SG proteomes studies [40,41,42,43,44]. More than 80% of SG proteomes have a protein binding function, implying that SG component candidates require specific binding to an SG-entry protein, such as G3BPs, in order to be sequestered into SGs. The second abundant term is RNA binding function, which also supports the underlying mechanisms of protein entry into SGs. Cadherin binding proteins are found in SG proteomes. The role of these proteins is still elusive, but it is likely that they would be involved in the movement or transport of SGs.

**Figure 4 ijms-25-12950-f004:**
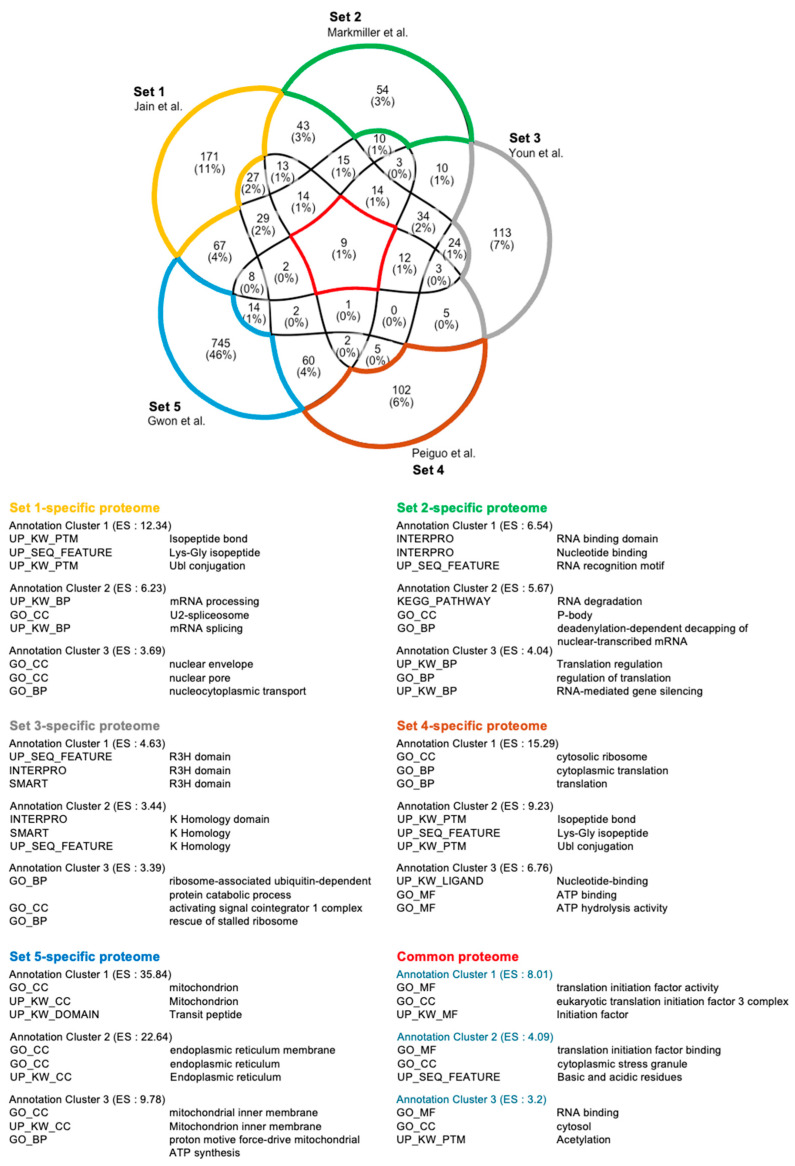
Functional cluster analysis of SGs. The five SG proteomes contain dynamic components [40,41,42,43,44]. These SG proteomes have only nine proteins in common, in which the functional annotation cluster of nine proteins well reflects the typical function of SGs. The number of each proteome-specific protein is higher than common proteins, suggesting that cell type, stressors, and various factors have a great influence on the SG components.

**Figure 5 ijms-25-12950-f005:**
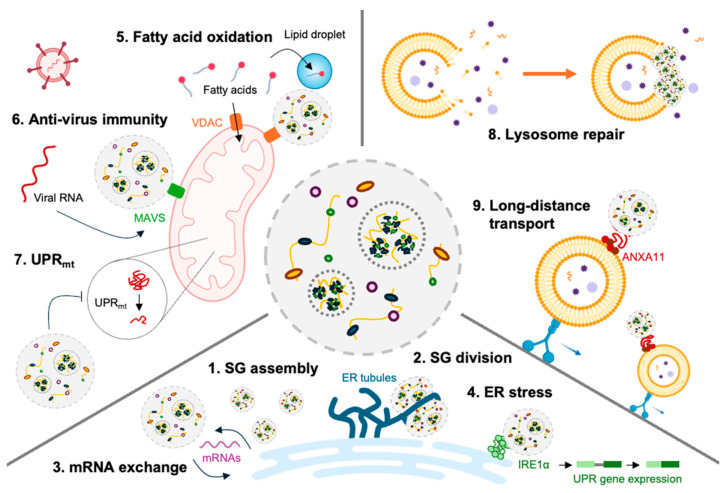
Interaction between SGs and organelles. SGs regulate cell activity through interactions with organelles such as ER, mitochondria, and lysosomes. (1) ER provides a working hub where SGs can be assembled. (2) ER tubule surrounds SGs to induce division. (3) ER determines the fate of mRNA by exchanging mRNA with the SGs. (4) SGs regulate the degree of ER stress by inducing IRE1α clustering. (5) SGs inhibit fat acid oxidization by redirecting fatty acids to lipid droplets. (6) SGs and MAVS on mitochondrial membranes bind to induce anti-virus activity. (7) The activity of mitochondrial UPR induces the formation of SGs, and SGs inhibit normal UPR activity. (8) The damaged lysosomal membrane is restored and stabilized by plugged SGs. (9) SGs hitchhike on lysosomes for long-distance travel to the distal part of the cells.

**Table 1 ijms-25-12950-t001:** Immune-related functional annotation clustering of SG proteome.

Enrichment Score (ES)	Term		Counts	Reference SG Proteome
Annotation Cluster 21 (ES = 3.57)	GO:BP	Innate immune response	30	[44]
	GO:BP	Defense response to virus	18	
	KW:BP	Innate immunity	28	
	KW:BP	Antiviral defense	14	
	KW:BP	Immunity	30	
Annotation Cluster 12 (ES = 5.59)	GO:BP	Defense response to virus	17	[40]
	KW:BP	Antiviral defense	14	
	GO:BP	Innate immune response	23	
	KW:BP	Innate immunity	23	
	KW:BP	immunity	24	
Annotation Cluster 28 (ES = 2.00)	GO:BP	Defense response to virus	10	[41]
	KW:BP	Antiviral defense	8	
	GO:BP	Innate immune response	15	
	KW:BP	Innate immunity	14	
	KW:BP	Immunity	15	
Annotation Cluster 22 (ES = 3.08)	GO:BP	DNA duplex unwinding	9	[42]
	KW:BP	Innate immunity	18	
	GO:BP	Innate immune response	16	
	KW:BP	Immunity	19	
Annotation Cluster 133 (ES = 0.13)	KW:BP	Innate immunity	20	[43]
	GO:BP	Innate immune response	20	
	KW:BP	Immunity	21	

## Data Availability

Not applicable.

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
