# Peer review of "Stress Granules in Infectious Disease: Cellular Principles and Dynamic Roles in Immunity and Organelles"

_ijms, 2024, doi:10.3390/ijms252312950_

Round 1
Reviewer 1 Report
Comments and Suggestions for Authors
In this review, Jaewhan Kim and Chang-Hwa Song provide a comprehensive summary of the role of stress granules (SGs) in immunity and cellular organelles within the context of infectious diseases. The authors critically analyze the regulatory functions of SGs in host cells during infections. Although the manuscript exhibits a clear logical structure, it lacks in-depth insights into the subject matter. The current manuscript contains several issues that require improvement and enhancement.
1. Figures 1 to 3 present only five existing studies and lack comprehensive analysis and visualization.
2. Figure 4 presents extensive content; however, it lacks a comprehensive summary of the information, indicating a need for further enhancement.
3. The scope of the title is excessively broad and lacks focus; therefore, it requires revision.
4. The fifth section should serve as the focal point of the article; however, the author addresses only a limited number of viruses and bacteria. It is suggested that the discussion be organized by mechanisms and that additional content be summarized in tables.
5. The discussion of future research trends in Section 6 is not deep enough and needs to be strengthened.
6. The content from lines 50 to 56 is highly repetitive with the content from lines 57 to 63; please revise it.
7. Please verify that the references comply with the journal's formatting requirements, as there are inconsistencies.
Author Response
|
Comments 1: Figures 1 to 3 present only five existing studies and lack comprehensive analysis and visualization. |
|
Response 1: As suggested by the reviewer, we have addressed comprehensive analysis of five existing studies of SG proteome and added a new figure (p. 4~6, line 148~176; revised Figure 4). |
|
Comments 2: Figure 4 presents extensive content; however, it lacks a comprehensive summary of the information, indicating a need for further enhancement. |
|
Response 2: We apologize for the lack of comprehensive summary of Figure 4 in original manuscript. We have addressed more detailed information in revised manuscript (p. 6~8, line 206~235). |
|
Comments 3: The scope of the title is excessively broad and lacks focus; therefore, it requires revision. |
|
Response 3: As commented by the reviewer, we have changed the title of this review paper (p. 1, line 2~3). |
|
Comments 4: The fifth section should serve as the focal point of the article; however, the author addresses only a limited number of viruses and bacteria. It is suggested that the discussion be organized by mechanisms and that additional content be summarized in tables. |
|
Response 4: We appreciate the reviewer’s constructive comment about fifth section. As suggested by the reviewer, we have added additional summary table including their regulation mechanism of SGs and outcome (revised Table 2). |
|
Comments 5: The discussion of future research trends in Section 6 is not deep enough and needs to be strengthened. |
|
Response 5: We apologize for insufficient prospects and future directions. We have added more informative and in-depth discussion for prospects and further directions (p. 16, line 563~605). |
|
Comments 6: The content from lines 50 to 56 is highly repetitive with the content from lines 57 to 63; please revise it. |
|
Response 6: As commented by the reviewer, we have deleted repetitive paragraph (line 50~56, in original manuscript). |
|
Comments 7: Please verify that the references comply with the journal's formatting requirements, as there are inconsistencies. |
Reviewer 2 Report
Comments and Suggestions for Authors
The battle against infectious diseases has been a long-standing challenge for humanity. Dr. Jaehwan Kim and Professor Changhwa Song have significantly contributed to readers' understanding of the relationship between stress granules (SGs) and infectious diseases through this review paper, which holds immense academic value. However, certain details require refinement, and I would like to offer several essential suggestions for improvement:
1. Introduction Part: The patterns of SG formation in viral and bacterial diseases should be specified and elaborated upon.
2. Introduction Part: Prospects for future disease control should be included to provide a forward-looking perspective.
3. Section 2: Additional explanations or diagrams are needed in the part discussing the composition and dynamics of SGs.
4. Table 1: The references supporting this table are somewhat insufficient and should be strengthened.
5. Figure 4: In the depiction, greater emphasis should be placed on aspects that are more striking and aligned with the paper's overall direction.
Author Response
|
Comments 1: Introduction Part: The patterns of SG formation in viral and bacterial diseases should be specified and elaborated upon. |
|
Response 1: We appreciate the reviewer’s comment on introduction part. We have addressed about the patterns of SG formation in viral and bacterial infection in revised manuscript (p. 2, line 51~62). |
|
Comments 2: Introduction Part: Prospects for future disease control should be included to provide a forward-looking perspective. |
|
Response 2: We have addressed more forward-looking perspective suggestion and discussion in revised manuscript (p. 2, line 59~62; p. 16, line 563~605). |
|
Comments 3: Section 2: Additional explanations or diagrams are needed in the part discussing the composition and dynamics of SGs. |
|
Response 3: As suggested by the reviewer, we have added more explanations and details on SG composition and dynamics in revised manuscript (p. 2, line 73~97). |
|
Comments 4: Table 1: The references supporting this table are somewhat insufficient and should be strengthened. |
|
Response 4: We apologize for the insufficient references in Table 1. However, as we re-analyzed the published SG proteomes from five papers to suggest that immune-associated proteins are included in SG proteomes, the references in Table 1 are appropriately added. To clarify the context and ensure the reference SG proteomes, we have changed “Reference” to “Reference SG proteome” in revised Table 1. In addition, to strengthen our table, we have added one more figure to give more detailed information about the five SG proteomes (revised Figure 4). |
|
Comments 5: Figure 4: In the depiction, greater emphasis should be placed on aspects that are more striking and aligned with the paper's overall direction. |
|
Response 5: As suggested by the reviewer, we have added one paragraph which briefly explain Figure 4 in original manuscript (Figure 5 in revised manuscript) and emphasize the importance of the communication between SGs and organelles in revised manuscript (p. 6~8, line 206~235). |
Round 2
Reviewer 1 Report
Comments and Suggestions for Authors
The authors have addressed all concerns and I recommend publication of this Review.